# Hierarchical Optimal Transport
# for Document Representation

**Mikhail Yurochkin**[1,3]
mikhail.yurochkin@ibm.com

**Sebastian Claici**[2,3]
sclaici@mit.edu

**Edward Chien**[2,3]
edchien@mit.edu

**Farzaneh Mirzazadeh**[1,3]
farzaneh@ibm.com

**Justin Solomon**[2,3]
jsolomon@mit.edu

IBM Research,[1] MIT CSAIL,[2] MIT-IBM Watson AI Lab[3]

## Abstract

The ability to measure similarity between documents enables intelligent summarization and analysis of large corpora. Past distances between documents suffer from either an inability to incorporate semantic similarities between words or from scalability issues. As an alternative, we introduce hierarchical optimal transport as a *meta*-distance between documents, where documents are modeled as distributions over topics, which themselves are modeled as distributions over words. We then solve an optimal transport problem on the smaller topic space to compute a similarity score. We give conditions on the topics under which this construction defines a distance, and we relate it to the word mover's distance. We evaluate our technique for $k$-NN classification and show better interpretability and scalability with comparable performance to current methods at a fraction of the cost.[1]

## 1 Introduction

Topic models like latent Dirichlet allocation (LDA) (Blei et al., 2003) are major workhorses for summarizing document collections. Typically, a topic model represents topics as distributions over the vocabulary (i.e., unique words in the corpus); documents are then modeled as distributions over topics. In this approach, words are vertices of a simplex whose dimension equals the vocabulary size and for which the distance between any pair of words is the same. More recently, word embeddings map words into high-dimensional space such that co-occurring words tend to be closer to each other than unrelated words (Mikolov et al., 2013; Pennington et al., 2014). Kusner et al. (2015) combine the geometry of word embedding space with optimal transport to propose the *word mover's distance* (WMD), a powerful document distance metric limited mostly by computational complexity.

As an alternative to WMD, in this paper we combine hierarchical latent structures from topic models with geometry from word embeddings. We propose *hierarchical* optimal topic transport (HOTT) document distances, which combine language information from word embeddings with corpus-specific, semantically-meaningful topic distributions from latent Dirichlet allocation (LDA) (Blei et al., 2003). This document distance is more efficient and more interpretable than WMD.

We give conditions under which HOTT gives a metric and show how it relates to WMD. We test against existing metrics on $k$-NN classification and show that it outperforms others on average. It performs especially well on corpora with longer documents and is robust to the number of topics and word embedding quality. Additionally, we consider two applications requiring pairwise distances.

The first is visualization of the metric with t-SNE (van der Maaten & Hinton, 2008). The second is link prediction from a citation network, cast as pairwise classification using HOTT features.

**Contributions.** We introduce *hierarchical* optimal transport to measure dissimilarities between distributions with common structure. We apply our method to document classification, where topics from a topic modeler represent the shared structure. Our approach

- is **computationally efficient**, since HOTT distances involve transport with small numbers of sites;
- uses corpus-specific topic and document distributions, providing **higher-level interpretability**;
- has **comparable performance** to WMD and other baselines for $k$-NN classification; and
- is **practical** in applications where all pairwise document distances are needed.

## 2 Related work

Document representation and similarity assessment are key applications in learning. Many methods are based on the bag-of-words (BOW), which represents documents as vectors in $\mathbb{R}^{|V|}$, where $|V|$ is the vocabulary size; each coordinate equals the number of times a word appears. Other weightings include term frequency inverse document frequency (TF-IDF) (Luhn, 1957; Spärck Jones, 1972) and latent semantic indexing (LSI) (Deerwester et al., 1990). Latent Dirichlet allocation (LDA) (Blei et al., 2003) is a hierarchical Bayesian model where documents are represented as admixtures of latent topics and admixture weights provide low-dimensional representations. These representations equipped with the $l_2$ metric comprise early examples of document dissimilarity scores.

Recent document distances employ more sophisticated methods. WMD incorporates word embeddings to account for word similarities (Kusner et al., 2015) (see §3). Huang et al. (2016) extend WMD to the supervised setting, modifying embeddings so that documents in the same class are close and documents from different classes are far. Due to computational complexity, these approaches are impractical for large corpora or documents with many unique words.

Wu & Li (2017) attempt to address the complexity of WMD via a topic mover's distance (TMD). While their $k$-NN classification results are comparable to WMD, they use significantly more topics, generated with a Poisson infinite relational model. This reduces semantic content and interpretability, with less significant computational speedup. They also do not leverage language information from word embeddings or otherwise. Xu et al. (2018) jointly learn topics and word embeddings, limiting the complexity to under a hundred words, which is not suited for natural language processing.

Wu et al. (2018) approximate WMD using a random feature kernel. In their method, the WMD from corpus documents to a selection of random short documents facilitates approximation of pairwise WMD. The resulting word mover's embedding (WME) has similar performance with significant speedups. Their method, however, requires parameter tuning in selecting the random document set and lacks topic-level interpretability. Additionally, they do not show full-metric applications. Lastly, Wan (2007), whose work predates (Kusner et al., 2015), applies transport to blocks of text.

## 3 Background

**Discrete optimal transport.** Optimal transport (OT) is a rich theory; we only need a small part and refer the reader to (Villani, 2009; Santambrogio, 2015) for mathematical foundations and to (Peyré & Cuturi, 2018; Solomon, 2018) for applications. Here, we focus on discrete-to-discrete OT.

Let $\mathbf{x} = \{x_1, \dots, x_n\}$ and $\mathbf{y} = \{y_1, \dots, y_m\}$ be two sets of points (sites) in a metric space. Let $\Delta^n \subset \mathbb{R}^{n+1}$ denote the probability simplex on $n$ elements, and let $p \in \Delta^n$ and $q \in \Delta^m$ be distributions over $\mathbf{x}$ and $\mathbf{y}$. Then, the 1-Wasserstein distance between $p$ and $q$ is

$$W_1(p, q) = \begin{cases} \min_{\Gamma \in \mathbb{R}_+^{n \times m}} & \sum_{i,j} C_{i,j} \Gamma_{i,j} \\ \text{subject to} & \sum_j \Gamma_{i,j} = p_i \text{ and } \sum_i \Gamma_{i,j} = q_j, \end{cases} \tag{1}$$

where the cost matrix $C$ has entries $C_{i,j} = d(x_i, y_j)$, where $d(\cdot, \cdot)$ denotes the distance. The constraints allow $\Gamma$ to be interpreted as a transport plan or matching between $p$ and $q$. The linear program (1) can be solved using the Hungarian algorithm (Kuhn, 1955), with complexity $O(l^3 \log l)$ where $l = \max(n, m)$. While entropic regularization can accelerate OT in learning environments (Cuturi, 2013), it is most successful when the support of the distributions is large as it has complexity

$O(l^2/\varepsilon^2)$. In our case, the number of topics in each document is small, and the linear program is typically faster if we need an accurate solution (i.e. if $\varepsilon$ is small).

**Word mover's distance.** Given an embedding of a vocabulary as $V \subset \mathbb{R}^n$, the Euclidean metric puts a geometry on the words in $V$. A corpus $D = \{d^1, d^2, \ldots d^{|D|}\}$ can be represented using distributions over $V$ via a normalized BOW. In particular, $d^i \in \Delta^{l_i}$, where $l_i$ is the number of unique words in a document $d^i$, and $d^i_j = c^i_j/|d^i|$, where $c^i_j$ is the count of word $v_j$ in $d^i$ and $|d^i|$ is the number of words in $d^i$. The WMD between documents $d^1$ and $d^2$ is then $WMD(d^1, d^2) = W_1(d^1, d^2)$.

The complexity of computing WMD depends heavily on $l = \max(l_1, l_2)$; for longer documents, $l$ may be a significant fraction of $|V|$. To evaluate the full metric on a corpus, the complexity is $O(|D|^2 l^3 \log l)$, since $WMD$ must be computed pairwise. Kusner et al. (2015) test WMD for $k$-NN classification. To circumvent complexity issues, they introduce a pruning procedure using a relaxed word mover's distance (RWMD) to lower-bound WMD. On the larger 20NEWS dataset, they additionally remove infrequent words by using only the top 500 words to generate a representation.

# 4  Hierarchical optimal transport

Assume a topic model produces corpus-specific topics $T = \{t_1, t_2, \ldots, t_{|T|}\} \subset \Delta^{|V|}$, which are distributions over words, as well as document distributions $\bar{d}^i \in \Delta^{|T|}$ over topics. WMD defines a metric $WMD(t_i, t_j)$ between topics; we consider discrete transport over $T$ as a metric space.

We define the hierarchical topic transport distance (HOTT) between documents $d^1$ and $d^2$ as

$$HOTT(d^1, d^2) = W_1 \left( \sum_{k=1}^{|T|} \bar{d}^1_k \delta_{t_k}, \sum_{k=1}^{|T|} \bar{d}^2_k \delta_{t_k} \right),$$

where each Dirac delta $\delta_{t_k}$ is a probability distribution supported on the corresponding topic $t_k$ and where the ground metric is WMD between topics as distributions over words. The resulting transport problem leverages topic correspondences provided by WMD in the base metric. This explains the *hierarchical* nature of our proposed distance.

Our construction uses transport *twice*: WMD provides topic distances, which are subsequently the costs in the HOTT problem. This hierarchical structure greatly reduces runtime, since $|T| \ll l$; the costs for HOTT can be precomputed once per corpus. The expense of evaluating pairwise distances is drastically lower, since pairwise distances between topics may be precomputed and stored. Even as document length and corpus size increase, the transport problem for HOTT remains the same size. Hence, full metric computations are feasible on larger datasets with longer documents.

When computing $WMD(t_i, t_j)$, we reduce computational time by truncating topics to a small amount of words carrying the majority of the topic mass and re-normalize. This procedure is motivated by interpretability considerations and estimation variance of the tail probabilities. On the interpretability side, LDA topics are often displayed using a few dozen top words, providing a human-understandable tag. Semantic coherence, a popular topic modeling evaluation metric, also is based on heavily-weighted words and was demonstrated to align with human evaluation of topic models (Newman et al., 2010). Moreover, any topic modeling inference procedure, e.g. Gibbs sampling (Griffiths & Steyvers, 2004), has estimation variance that may dominate tail probabilities, making them unreliable. Hence, we truncate to the top 20 words when computing WMD between topics. We empirically verify that truncation to any small number of words performs equally well in §5.3.

In topic models, documents are assumed to be represented by a small subset of topics of size $\kappa_i \ll |T|$ (e.g., in Figure 1, *books* are majorly described by three topics), but in practice document topic proportions tend to be dense with little mass outside of the dominant topics. Williamson et al. (2010) propose an LDA extension enforcing sparsity of the topic proportions, at the cost of slower inference. When computing HOTT, we simply truncate LDA topic proportions at $1/(|T| + 1)$, the value below LDA's uniform topic proportion prior, and re-normalize. This reduces complexity of our approach without performance loss as we show empirically in §5.2 and §5.3.

**Metric properties of HOTT.** If each document can be uniquely represented as a linear combination of topics $d^i = \sum_{k=1}^{|T|} \bar{d}^i_k t_k$, and each topic is unique, then $HOTT$ is a metric on document space. We present a brief proof in the supplementary material.

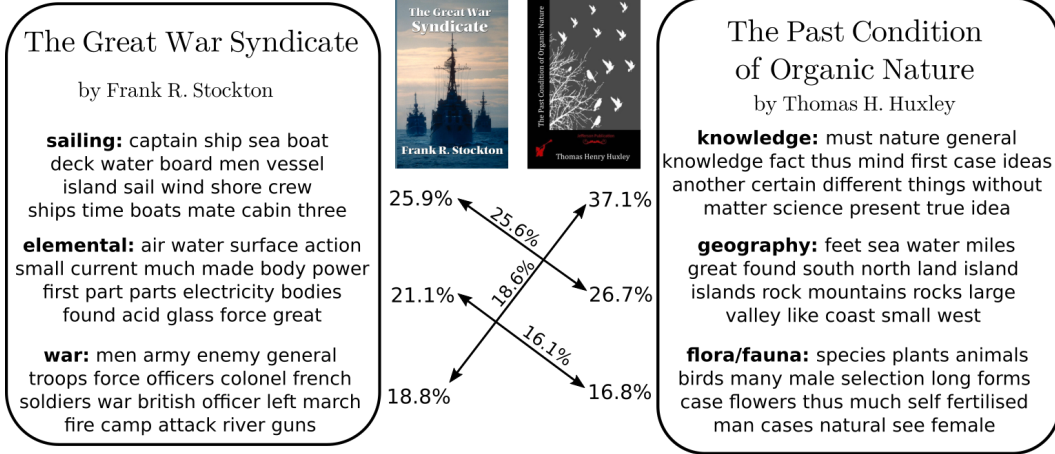

Figure 1: Topic transport interpretability. We show two books from GUTENBERG and their heaviest-weighted topics (bolded topic names are manually assigned). The first involves steamship warfare, while the second involves biology. Left and right column percentages indicate the weights of the topics in the corresponding texts. Percentages labeling the arrows indicate the transported mass between the corresponding topics, which match semantically-similar topics.

**Topic-level interpretability.** The additional level of abstraction promotes higher-level interpretability at the level of topics as opposed to dense word-level correspondences from WMD. We provide an example in Figure 1. This diagram illustrates two books from the GUTENBERG dataset and the semantically meaningful transport between their three most heavily-weighted topics. Remaining topics and less prominent transport terms account for the remainder of the transport plan not illustrated.

**Relation to WMD.** First we note that if $|T| = |V|$ and topics consist of single words covering the vocabulary, then HOTT becomes WMD. In well-behaved topic models, this is expected as $|T| \to |V|$. Allowing $|T|$ to vary produces different levels of granularity for our topics as well as a trade-off between computational speed and topic specificity. When $|T| \ll |V|$, we argue that WMD is upper bounded by HOTT and two terms that represent topic modeling loss. By the triangle inequality,

$$WMD(d^i, d^j) \leq W_1\left(d^i, \sum_{k=1}^{|T|} \bar{d}_k^i t_k\right) + W_1\left(\sum_{k=1}^{|T|} \bar{d}_k^i t_k, \sum_{k=1}^{|T|} \bar{d}_k^j t_k\right) + W_1\left(\sum_{k=1}^{|T|} \bar{d}_k^j t_k, d^j\right). \quad (2)$$

LDA inference minimizes $\mathrm{KL}(d^i \| \sum_{k=1}^{|T|} \bar{d}_k^i t_k)$ over topic proportions $\bar{d}^i$ for a given document $d^i$; hence, we look to relate Kullback–Leibler divergence to $W_1$. In finite-diameter metric spaces, $W_1(\mu, \nu) \leq \mathrm{diam}(X)\sqrt{\frac{1}{2}\mathrm{KL}(\mu\|\nu)}$, which follows from inequalities relating Wasserstein distances to KL divergence (Otto & Villani, 2000). The middle term satisfies the following inequality:

$$W_1\left(\sum_{k=1}^{|T|} \bar{d}_k^i t_k, \sum_{k=1}^{|T|} \bar{d}_k^j t_k\right) \leq W_1\left(\sum_{k=1}^{|T|} \bar{d}_k^i \delta_{t_k}, \sum_{k=1}^{|T|} \bar{d}_k^j \delta_{t_k}\right), \quad (3)$$

where on the right we have $HOTT(d^1, d^2)$. The optimal topic transport on the right implies an equal-cost transport of the corresponding linear combinations of topic distributions on the left. The inequality follows since $W_1$ gives the *optimal* transport cost. Combining into a single inequality,

$$WMD(d^i, d^j) \leq HOTT(d^i, d^j) + \mathrm{diam}(X)\left[\sqrt{\frac{1}{2}\mathrm{KL}\left(d^j \left\| \sum_{k=1}^{|T|} \bar{d}_k^j t_k\right.\right)} + \sqrt{\frac{1}{2}\mathrm{KL}\left(d^i \left\| \sum_{k=1}^{|T|} \bar{d}_k^i t_k\right.\right)}\right].$$

WMD involves a large transport problem and Kusner et al. (2015) propose relaxed WMD (RWMD), a relaxation via a lower bound (see also Atasu & Mittelholzer (2019) for a GPU-accelerated variant). We next show that RWMD is not always a good lower bound on WMD.

**RWMD–Hausdorff bound.** Consider the optimization in (1) for calculating $WMD(d^1, d^2)$, and remove the marginal constraint on $d^2$. The resulting optimal $\Gamma$ is no longer a transport plan, but rather moves mass on words in $d^1$ to their nearest words in $d^2$, only considering the support of $d^2$ and not its density values. Removing the marginal constraint on $d^1$ produces symmetric behavior; $RWMD(d^1, d^2)$ is defined to be the larger cost of these relaxed problems.

Suppose that word $v_j$ is shared by $d^1$ and $d^2$. Then, the mass on $v_j$ in $d^1$ and $d^2$ in each relaxed problems will not move and contributes zero cost. In the worst case, if $d^1$ and $d^2$ contain the same words, i.e., $\text{supp}(d^1) = \text{supp}(d^2)$, then $RWMD(d^1, d^2) = 0$. More generally, the closer the supports of two documents (over $V$), the looser RWMD might be as a lower bound.

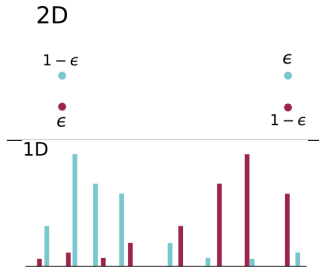

Figure 2 illustrates two examples. In the 2D example, $1 - \epsilon$ and $\epsilon$ denote the masses in the teal and maroon documents. The 1D example uses histograms to represent masses in the two documents. In both, RWMD is nearly zero as masses do not have far to move, while the WMD will be larger thanks to the constraints.

To make this precise we provide the following tight upper bound: $RWMD(d^1, d^2) \leq d_H(\text{supp}(d^1), \text{supp}(d^2))$, the Hausdorff distance between the supports of $d^1$ and $d^2$. Let $X = \text{supp}(d^1)$ and $Y = \text{supp}(d^2)$; and let $RWMD_1(d^1, d^2)$ and $RWMD_2(d^1, d^2)$ denote the relaxed optimal values when the marginal constraints on $d^1$ and $d^2$ are kept, respectively:

Figure 2: RWMD as a poor approximation to WMD

$$d_H(X, Y) = \max \left( \sup_{x \in X} \inf_{y \in Y} d(x, y), \sup_{y \in Y} \inf_{x \in X} d(x, y) \right)$$
$$\geq \max \left( RWMD_1(d^1, d^2), RWMD_2(d^1, d^2) \right) = RWMD(d^1, d^2).$$

The inequality follows since the left argument of the max is the furthest mass must travel in the solution to $RWMD_1$, while the right is the furthest mass must travel in the solution to $RWMD_2$. It is tight if the documents have singleton support and whenever $d^1$ and $d^2$ are supported on parallel affine subspaces and are translates in a normal direction. A 2D example is in Figure 2.

The preceding discussion suggests that RWMD is not an appropriate way to speed up WMD for long documents with overlapping support, scenario where WMD computational complexity is especially prohibitive. The GUTENBERG dataset showcases this failure, in which documents frequently have common words. Our proposed HOTT does not suffer from this failure mode, while being significantly faster and as accurate as WMD. We verify this in the subsequent experimental studies. In the supplementary materials we present a brief empirical analysis relating HOTT and RWMD to WMD in terms of Mantel correlation and a Frobenius norm.

## 5   Experiments

We present timings for metric computation and consider applications where distance between documents plays a crucial role: $k$-NN classification, low-dimensional visualization, and link prediction.

### 5.1   Computational timings

**HOTT implementation.** During training, we fit LDA with 70 topics using a Gibbs sampler (Griffiths & Steyvers, 2004). Topics are truncated to the 20 most heavily-weighted words and renormalized. The pairwise distances between topics $WMD(t_i, t_j)$ are precomputed with words embedded in $\mathbb{R}^{300}$ using *GloVe* (Pennington et al., 2014). To evaluate HOTT at testing time, a few iterations of the Gibbs sampler are run to obtain topic proportions $\bar{d}^i$ of a new document $d^i$. When computing HOTT between a pair of documents we truncate topic proportions at $1/(|T| + 1)$ and renormalize. Every instance of the OT linear program is solved using Gurobi (Gurobi Optimization, 2018).

We note that LDA inference may be carried out using any other approaches, e.g. stochastic/streaming variational inference (Hoffman et al., 2013; Broderick et al., 2013) or geometric algorithms (Yurochkin & Nguyen, 2016; Yurochkin et al., 2019). We chose the MCMC variant (Griffiths & Steyvers, 2004) for its strong theoretical guarantees, simplicity and wide adoption in the topic modeling literature.

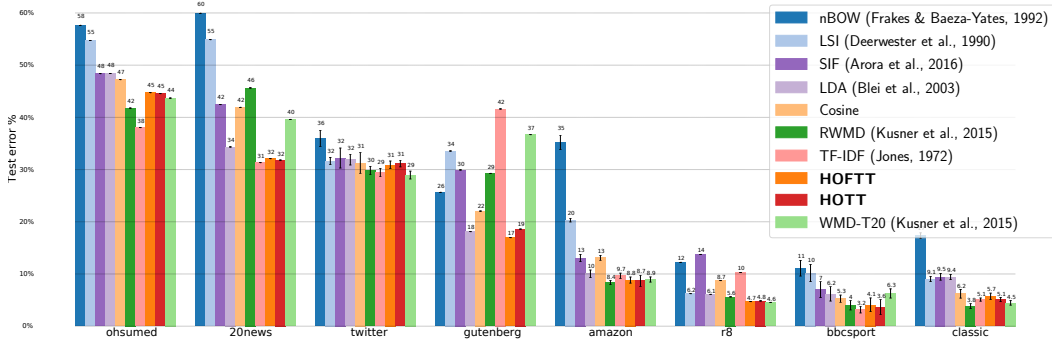

Figure 3: $k$-NN classification performance across datasets

Table 1: Dataset statistics and document pairs per second; higher is better. HOTT has higher throughput and excels on long documents with large portions of the vocabulary (as in GUTENBERG).

| | | DATASET STATISTICS | | | | | | PAIRS PER SECOND | | | |
|---|---|---|---|---|---|---|---|---|---|---|---|
| DATASET | $|D|$ | $|V|$ | IOU | AVG($l$) | AVG($\kappa$) | CLASSES | RWMD | WMD | WMDT20 | HOFTT | HOTT |
| BBCSPORT | 737 | 3657 | 0.066 | 116.5 | 11.7 | 5 | 1494 | 526 | 1545 | 2016 | **2548** |
| TWITTER | 3108 | 1205 | 0.029 | 9.7 | 6.3 | 3 | **2664** | 2536 | 2194 | 1384 | 1552 |
| OHSUMED | 9152 | 8261 | 0.046 | 59.4 | 11.0 | 10 | 454 | 377 | 473 | 829 | **908** |
| CLASSIC | 7093 | 5813 | 0.017 | 38.5 | 8.7 | 4 | 816 | 689 | 720 | 980 | **1053** |
| REUTERS8 | 7674 | 5495 | 0.06 | 35.7 | 8.7 | 8 | 834 | 685 | 672 | 918 | **989** |
| AMAZON | 8000 | 16753 | 0.019 | 44.3 | 9.0 | 4 | 289 | 259 | 253 | 927 | **966** |
| 20NEWS | 13277 | 9251 | 0.011 | 69.3 | 10.5 | 20 | 338 | 260 | 384 | 652 | **699** |
| GUTENBERG | 3037 | 15000 | 0.25 | 4367 | 13.3 | 142 | 2 | 0.3 | 359 | 1503 | **1720** |

**Topic computations.** The preprocessing steps of our method—computing LDA topics and the topic to topic pairwise distance matrix—are dwarfed by the cost of computing the full document-to-document pairwise distance matrix. The complexity of base metric computation in our implementation is $O(|T|^2)$, since $|\text{supp}(t_i)| = 20$ for all topics, leading to a relatively small OT instance.

**HOTT computations.** All distance computations were implemented in Python 3.7 and run on an Intel i7-6700K at 4GHz with 32GB of RAM. Timings for pairwise distance computations are in Table 1 (right). HOTT outperforms RWMD and WMD in terms of speed as it solves a significantly smaller linear program. On the left side of Table 1 we summarize relevant dataset statistics: $|D|$ is the number of documents; $|V|$ is the vocabulary size; intersection over union (IOU) characterizes average overlap in words between pairs of documents; AVG($l$) is the average number of unique words per document and AVG($\kappa$) is the average number of major topics (i.e., after truncation) per document.

## 5.2  $k$-NN classification

We follow the setup of Kusner et al. (2015) to evaluate performance of HOTT on $k$-NN classification.

**Datasets.** We consider 8 document classification datasets: BBC sports news articles (BBCSPORT) labeled by sport; tweets labeled by sentiments (TWITTER) (Sanders, 2011); Amazon reviews labeled by category (AMAZON); Reuters news articles labeled by topic (REUTERS) (we use the 8-class version and train-test split of Cachopo et al. (2007)); medical abstracts labeled by cardiovascular disease types (OHSUMED) (using 10 classes and train-test split as in Kusner et al. (2015)); sentences from scientific articles labeled by publisher (CLASSIC); newsgroup posts labeled by category (20NEWS), with "by-date" train-test split and removing headers, footers and quotes;[2] and Project Gutenberg full-length books from 142 authors (GUTENBERG) using the author names as classes and 80/20 train-test split in the order of document appearance. For GUTENBERG, we reduced the vocabulary to the most common 15000 words. For 20NEWS, we removed words appearing in $\leq 5$ documents.

**Baselines.** We focus on evaluating HOTT and a variation without topic proportion truncation (HOFTT: hierarchical optimal full topic transport) as alternatives to RWMD in a variety of metric-dependent tasks. As demonstrated by the authors, RWMD has nearly identical performance to WMD, while being more computationally feasible. Additionally, we analyze a naïve approach for speeding-up WMD where we truncate documents to their top 20 unique words (WMD-T20), making complexity comparable to HOTT (yet $20 > $AVG$(\kappa)$ on all datasets). For $k$-NN classification, we also consider baselines that represent documents in vector form and use Euclidean distances: normalized bag-of-words (nBOW) (Frakes & Baeza-Yates, 1992); latent semantic indexing (LSI) (Deerwester et al., 1990); latent Dirichlet allocation (LDA) (Blei et al., 2003) trained with a Gibbs sampler (Griffiths & Steyvers, 2004); and term frequency inverse document frequency (TF-IDF) (Spärck Jones, 1972). We omit comparison to embedding via BOW weighted averaging as it was shown to be inferior to RWMD by Kusner et al. (2015) (i.e., Word Centroid Distance) and instead consider smooth inverse frequency (SIF), a recent document embedding method by Arora et al. (2016). We also compare to bag-of-words, where neighbors are identified using cosine similarity (Cosine). We use same pre-trained *GloVe* embeddings for HOTT, RWMD, SIF and truncated WMD and set the same number of topics $|T| = 70$ for HOTT, LDA and LSI; we provide experiments testing parameter sensitivity.

**Results.** We evaluate each method on $k$-NN classification (Fig. 3). There is no uniformly best method, but HOTT performs best on average (Fig. 4) We highlight the performance on the GUTENBERG dataset compared to RWMD. We anticipate poor performance of RWMD on GUTENBERG, since books contain more words, which can make RWMD degenerate (see §4 and Fig. 2). Also note strong performance of TF-IDF on OHSUMED and 20NEWS, which differs from results of Kusner et al. (2015). We believe this is due to a different normalization scheme. We used *TfidfTransformer* from scikit-learn (Pedregosa et al., 2011) with default settings. We conclude that HOTT is most powerful, both computationally (Table 1 right) and as a distance metric for $k$-NN classification (Figures 3 and 4), on larger corpora of longer documents, whereas on shorter documents both RWMD and HOTT perform similarly.

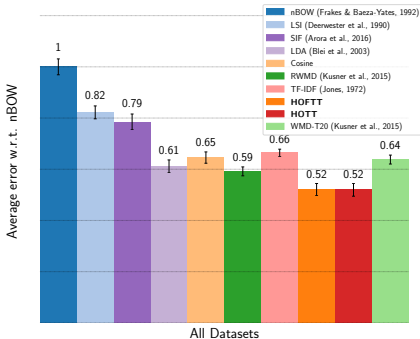

Figure 4: Aggregated $k$-NN classification performance normalized by nBOW

Another interesting observation is the effect of truncation: HOTT performs as well as HOFTT, meaning that truncating topic proportions of LDA does not prevent us from obtaining high-quality document distances in less computational time, whereas truncating unique words for WMD degrades its performance. This observation emphasizes the challenge of speeding up WMD, i.e. WMD *cannot* be made computationally efficient using truncation without degrading its performance. WMD-T20 is slower than HOTT (Table 1) and performs noticeably worse (Figure 4). Truncating WMD further will make its performance even worse, while truncating less will quickly lead to impractical run-time.

In the supplement, we complement our results considering 2-Wasserstein distance, and stemming, a popular text pre-processing procedure for topic models to reduce vocabulary size. HOTT continues to produce best performance on average. We restate that in all main text experiments we used 1-Wasserstein (i.e. eq. (1)) and did not stem, following experimental setup of Kusner et al. (2015).

## 5.3 Sensitivity analysis of HOTT

We analyze senstitivity of HOTT with respect to its components: word embeddings, number of LDA topics, and topic truncation level.

**Sensitivity to word embeddings.** We train *word2vec* (Mikolov et al., 2013) 200-dimensional embeddings on REUTERS and compare relevant methods with our default embedding (i.e., *GloVe*) and newly-trained *word2vec* embeddings. According to Mikolov et al. (2013), word embedding quality largely depends on data *quantity* rather than quality; hence we expect the performance to degrade. In Fig. 5a, RWMD and WMD truncated performances drop as expected, but HOTT and HOFTT remain stable; this behavior is likely due to the embedding-independent topic structure taken into consideration.

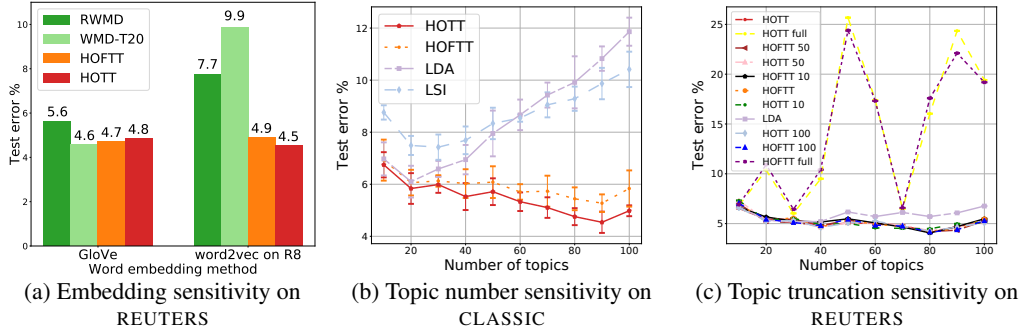

| (a) Embedding sensitivity on REUTERS | (b) Topic number sensitivity on CLASSIC | (c) Topic truncation sensitivity on REUTERS |
|---|---|---|

Figure 5: Sensitivity analysis: embedding, topic number and topic truncation

**Number of LDA topics.** In our experiments, we set $|T| = 70$. When the $|T|$ increases, LDA resembles the nBOW representation; correspondingly, HOTT approaches the WMD. The difference, however, is that nBOW is a weaker baseline, while WMD is powerful document distance. Using the CLASSIC dataset, in Fig. 5b we demonstrate that LDA (and LSI) may degrade with too many topics, while HOTT and HOFTT are robust to topic overparameterization. In this example, better performance of HOTT over HOFTT is likely due relatively short documents of the CLASSIC dataset.

While we have shown that HOTT is not sensitive to the choice of the number of topics, it is also possible to eliminate this parameter by using LDA inference algorithms that learn number of topics (Yurochkin et al., 2017) or adopting Bayesian nonparametric topic modes and corresponding inference schemes (Teh et al., 2006; Wang et al., 2011; Bryant & Sudderth, 2012).

**Topic truncation.** Fig. 5c demonstrates $k$-NN classification performance on the REUTERS dataset with varying topic truncation: top 10, 20 (HOTT and HOFTT), 50, 100 words and no truncation (HOTT full and HOFTT full); LDA performance is given for reference. Varying the truncation level does not affect the results significantly, however no truncation results in unstable performance.

## 5.4 t-SNE metric visualization

Visualizing metrics as point clouds provides useful qualitative information for human users. Unlike $k$-NN classification, most methods for this task require long-range distances and a full metric. Here, we use t-SNE (van der Maaten & Hinton, 2008) to visualize HOTT and RWMD on the CLASSIC dataset in Fig. 6. HOTT appears to more accurately separate the labeled points (color-coded). The supplementary material gives additional t-SNE results.

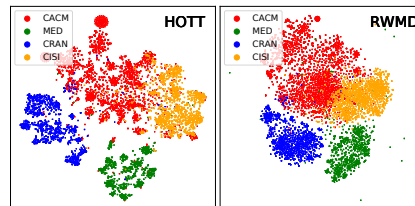

Figure 6: t-SNE on CLASSIC

## 5.5 Supervised link prediction

We next evaluate HOTT in a different prediction task: supervised link prediction on graphs defined on text domains, here *citation networks*. The specific task we address is the Kaggle challenge of Link Prediction TU.[3] In this challenge, a citation network is given as an undirected graph, where nodes are research papers and (undirected) edges represent citations. From this graph, edges have been removed at random. The task is to reconstruct the full network. The dataset contains 27770 papers (nodes). The training and testing sets consist of $615512$ and $32648$ node pairs (edges) respectively. For each paper, the available data only includes publication year, title, authors, and abstract.

To study the effectiveness of a distance-based model with HOTT for link prediction, we train a linear SVM classifier over the feature set $\Phi$, which includes the distance between the two abstracts $\phi_{dist}$ computed via one of {HOFT, HOTT, RWMD, WMD-T20}. For completeness, we also examine excluding the distance totally. We incrementally grow the feature sets $\Phi$ as: $\Phi_0 = \{\phi_{dist}\}$, $\Phi_1 = \{\phi_{dist}\} \cup \{\phi_1\}$, $\Phi_n = \{\phi_{dist}\} \cup \{\phi_1, \ldots, \phi_n\}$ where $\phi_1$ is the number of common words

Table 2: Link prediction: using distance (rows) for node-pair representations (cols).

| Distance | F1 Score | | | | |
|---|---|---|---|---|---|
| | $\Phi_0$ | $\Phi_1$ | $\Phi_2$ | $\Phi_3$ | $\Phi_4$ |
| HOFTT | **73.22** | **76.27** | **76.62** | **78.85** | **83.37** |
| HOTT | 73.19 | 76.03 | 76.24 | 78.64 | 83.25 |
| RWMD | 71.60 | 74.90 | 75.20 | 77.16 | 82.92 |
| WMD-T20 | 67.22 | 63.38 | 65.20 | 70.38 | 81.84 |
| None | — | 61.13 | 64.27 | 67.72 | 81.68 |

in the titles, $\phi_2$ the number of common authors, and $\phi_3$ and $\phi_4$ the signed and absolute difference between the publication years.

Table 2 presents the results; evaluation is based on the F1-Score. Consistently, HOFTT and HOTT are more effective than RWMD and WMD-T20 in all tests, and not using any of the distances consistently degrades the performance.

## 6 Conclusion

We have proposed a hierarchical method for comparing natural language documents that leverages optimal transport, topic modeling, and word embeddings. Specifically, word embeddings provide global semantic language information, while LDA topic models provide corpus-specific topics and topic distributions. Empirically these combine to give superior performance on various metric-based tasks. We hypothesize that modeling documents by their representative topics is better for highlighting differences despite the loss in resolution. HOTT appears to capture differences in the same way a person asked to compare two documents would: by breaking down each document into easy to understand concepts, and then comparing the concepts.

There are many avenues for future work. From a theoretical perspective, our use of a nested Wasserstein metric suggests further analysis of this hierarchical transport space. Insight gained in this direction may reveal the learning capacity of our method and inspire faster or more accurate algorithms. From a computational perspective, our approach currently combines word embeddings, topic models and OT, but these are all trained separately. End-to-end training that efficiently optimizes these three components jointly would likely improve performance and facilitate analysis of our algorithm as a unified approach to document comparison.

Finally, from an empirical perspective, the performance improvements we observe stem directly from a reduction in the size of the transport problem. Investigation of larger corpora with longer documents, and applications requiring the full set of pairwise distances are now feasible. We also can consider applications to modeling of images or 3D data.

**Acknowledgements.** J. Solomon acknowledges the generous support of Army Research Office grant W911NF1710068, Air Force Office of Scientific Research award FA9550-19-1-031, of National Science Foundation grant IIS-1838071, from an Amazon Research Award, from the MIT-IBM Watson AI Laboratory, from the Toyota-CSAIL Joint Research Center, from the QCRI–CSAIL Computer Science Research Program, and from a gift from Adobe Systems. Any opinions, findings, and conclusions or recommendations expressed in this material are those of the authors and do not necessarily reflect the views of these organizations.

## Footnotes

[1]Code: https://github.com/IBM/HOTT

[2]https://scikit-learn.org/0.19/datasets/twenty_newsgroups.html

[3] www.kaggle.com/c/link-prediction-tu

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
