[Supplementary Material]

# Supplementary material for Hierarchical Optimal Transport for Document Representation

**Mikhail Yurochkin**[1,3]
mikhail.yurochkin@ibm.com

**Sebastian Claici**[2,3]
sclaici@mit.edu

**Edward Chien**[2,3]
edchien@mit.edu

**Farzaneh Mirzazadeh**[1,3]
farzaneh@ibm.com

**Justin Solomon**[2,3]
jsolomon@mit.edu

IBM Research[1], MIT CSAIL[2], MIT-IBM Watson AI Lab[3]

## 1 Metric properties

*HOTT* is a metric in the lifted topic space since $W_p$ is a metric on distributions.

*Proof.* We can additionally prove that if we can exactly write $d^i = \sum_{k=1}^{|T|} \bar{d}_k^i t_k$ and if $t_i \neq t_j$ for $i \neq j$, then *HOTT* is a metric in document space.

Positivity, symmetry, and the triangle inequality follow from properties of $W_2$. We prove that if $HOTT(d^i, d^j) = 0$, then $d^i = d^j$. From the definition of *HOTT*,

$$HOTT(d^i, d^j) = W_2 \left( \sum_{k=1}^{|T|} \bar{d}_k^i \delta_{t_k}, \sum_{l=1}^{|T|} \bar{d}_l^j \delta_{t_l} \right).$$

If $HOTT(d^i, d^j) = 0$, then if the transport plan is positive at $T_{k,l}$, it must hold that $W_p(t_k, t_l) = 0$. Since $W_p$ is a metric on probability distributions, this implies $t_k = t_l$. As we assumed that topics are distinct, and that documents are uniquely represented as linear combinations of topics we have $d^i = d^j$. $\square$

## 2 HOTT/WMD/RWMD relation

Following the discussion in Section 4 of the main text, we relate HOTT and RWMD to WMD empirically in terms of Mantel correlation and a Frobenius norm. The results are in Table 1. While it is unsurprising that RWMD is more strongly correlated with WMD (HOTT is neither a lower nor an upper bound), we note that HOTT is on average a better approximation to WMD than RWMD.

## 3 Additional experimental results

In the main text, we used $W_1$ distance and did not do any vocabulary reduction, following the experimental setup of Kusner et al. (2015). $W_2$ distance has intuitive geometric properties and is equipped with a variety of theoretical characterizations (Villani, 2009); one intuition for the difference between $W_1$ and $W_2$ comes from an analogy to the differences between $l_1$ and $l_2$ regularization. On the other hand, stemming is a common vocabulary reduction technique to improve quality of topic models. Stemming attempts to merge terms which differ only in their ending, i.e. "cat" and "cats". As stemming sometimes produces words not available in the *GloVe* embeddings (Pennington et al.,

Table 1: Relation between the metrics. For each dataset, we compute distance matrices using exact WMD, RWMD, and HOTT from a few randomly-selected documents. We report results of a Mantel correlation test between WMD/HOTT and WMD/RWMD and the difference between cost matrices under a Frobenius norm.

| | Mantel | | $l_2$ | |
|---|---|---|---|---|
| Dataset | HOTT | RWMD | HOTT | RWMD |
| OHSUMED | 0.57 | 0.87 | 55 | 104 |
| 20NEWS | 0.62 | 0.90 | 90 | 99 |
| AMAZON | 0.49 | 0.84 | 70 | 65 |
| REUTERS | 0.72 | 0.91 | 130 | 151 |
| BBCSPORT | 0.76 | 0.92 | 28 | 90 |
| CLASSIC | 0.43 | 0.89 | 157 | 69 |
| Avg | 0.60 | 0.89 | 88 | 96 |

2014), to embed a stemmed word we take the average embeddings of the words mapped to it. We used *SnowballStemmer* available from the *nltk* Python package.

Figures 1 and 2 demonstrate results with $W_1$ and stemming; Figures 3 and 4 with $W_2$ and no stemming; Figures 5 and 6 with $W_2$ and stemming. In all settings HOTT and HOFTT are the best on average. Interestingly, using $W_2$ degrades performance of RWMD and WMD-T20, while our methods perform equally well with $W_1$ and $W_2$. Stemming tends to improve performance of nBOW, therefore aggregated results appear worse. Stemming also negatively effects RWMD and WMD-T20, while appears to have no effect on HOTT and HOFTT. For example, in the case of $W_2$ with stemming (Figures 5 and 6), RWMD is no longer superior to baselines LDA (Blei et al., 2003) and Cosine, while our methods maintain good performance. We conclude that our methods are more robust to the choice of text processing techniques and specifics of the Wasserstein distance.

In Figure 7 we present additional t-SNE (van der Maaten & Hinton, 2008) visualization results.

Figure 1: $W_1$ and stemming: $k$-NN classification performance across datasets

Figure 2: $W_1$ and stemming: $k$-NN classification performance normalized by nBOW

Figure 3: $W_2$ without stemming: $k$-NN classification performance across datasets

Figure 4: $W_2$ without stemming: aggregated $k$-NN classification performance normalized by nBOW

Figure 5: $W_2$ and stemming: $k$-NN classification performance across datasets

Figure 6: $W_2$ and stemming: aggregated $k$-NN classification performance normalized by nBOW

(a) 20NEWS

(b) AMAZON

(c) BBCSPORT

(d) OHSUMED

(e) REUTERS

(f) TWITTER

Figure 7: These are the additional t-SNE results on all other datasets, except GUTENBERG, which is excluded due to its high number of classes (142). These images show that clusters based on our metric better align with the labels, corresponding to a better metric than RWMD. Both methods perform poorly on TWITTER, a difficult dataset for topic modelling.