[Reviews · NeurIPS 2019]

Reviewer 1



Originality: The authors clearly distinguish their work from previous efforts in the related work section. TMD seems to be the most similar work, at least thematically, but was not included as a baseline. Quality: The results and proofs are thorough. Classification is run on a variety of datasets against reasonable baselines. The classification performance numbers are supplemented with sensitivity analysis, runtime information, and two additional tasks (t-SNE visualizations and link prediction). The proposed approaches work best on average across all datasets, but they only beat all baselines on 1 of the 8 datasets, a fact that could use more discussion in the paper. Clarity: The work is well-motivated, and the authors do a good job of anticipating reader questions. Some smaller points were unclear (see "Improvements"), but overall well-written. Significance: The good results on long texts (especially the full novels of Project Gutenberg) and focus on computational efficiency are especially promising as research continues to push towards using longer and more complex texts.

Reviewer 2



* Originality: The paper proposes a nice idea for speeding up WMD, using a hierarchy. This idea although was applied to several problems long time ago (e.g. hierarchical softmax) but it is novel for this problem. * Quality: The paper presents throughly properties of HOTT. Experiments showing that HOTT does work. * Clarity: The paper is NOT difficult to read to capture the main contributions. However, I did not check the math carefully. In the equation in line 95, what are deltas? * Significance: Although the main idea is nice, its implementation is heavily relied on pruning LDA's topics to make |T| and the number of words in each t small. I found it very tricky, and was wondering what if we also use the same heuristic for WMD.

Reviewer 3



Applying the idea of Word Mover's Distance to topic distribution representation of documents is interesting. While the overall idea is good, the evaluation is not good enough. Although WMD is far inefficient compared with the proposed method, its performance is better presented in the experiment. Instead, WMD-T20 is taken as a baseline. It is not clear why 20 is determined. While each of the topic is represented by top 20 words, the number of topics are decided as 70. So, other size of truncated WMD model should be compared. According to Figure 4 (a), WMD-T20 attains lower error than the proposed method on GloVe. It shows a quite different performance on word2vec. Why does this happen? If GloVe is used throughout the experiments, what results are obtained in Figure 5? In the definition of HOTT on page 3, why do you need to use delta-function? Since \overline{d^i} is defined as a distribution, a definition like W_1( \overline{d^1}, \overline{d^2} ) looks to be enough. Or, please make the formulas more clearly either they are distributions on topics or words.

[Author Response · NeurIPS 2019]

We thank the reviewers for their time, their valuable and encouraging feedback, and their recommendations for improvement. We remain confident that our work is of strong interest to the NeurIPS community and easily can incorporate the suggested changes in a revision for the conference. Answers to specific comments appear below.

**Interpretability**    To address **R2**'s concerns about interpretability, we refer to Figure 1 in the paper, where we show an example with two novels from the Gutenberg dataset. To interpret HOTT distance between a pair of documents, one simply needs to look at the slice of the transport map corresponding to the dominant topics (typically 3-4 per document) as we show in Figure 1. As a point of contrast, interpreting WMD would require investigating a transport map between all unique words in a document pair (thousands of unique words for books).

**R1** requested clarifications about Figure 1. R1 correctly interpreted the percentages in the figure and pointed out that topic titles are assigned by us, and not by any algorithm we use. We will improve the caption clarity accordingly.

**Theoretical contributions**    **R1** asked about the purpose of the RWMD-Hausdorff bound section. This section strengthens the motivation for our work: WMD has been shown to be successful, however may be too slow in practice; RWMD is a fast approximation empirically performing well in some cases, but we show that it may also be a very poor choice of metric in practical scenarios. To give an extreme example, the documents [good, bad, bad, ..., bad] and [bad, good, good, ..., good] are distance 0 from each other under RWMD. This motivates the study of alternative document distance metrics utilizing word embeddings geometry, which we do by proposing HOTT. We will make this connection more explicit in the paper.

**Technical details**    **R1** asked whether we can use *different cost metrics* to measure distance between topics. Our choice of Wasserstein metric is motivated by the improvement seen in using the Earth Mover's (Word Mover's) Distance as a document-to-document metric. Other distances between topics, i.e., Euclidean or cosine, do not exploit the geometry of word embeddings. We conjecture (and can easily verify that) they will perform worse.

**R2** and **R3** had questions about the definition of *HOTT* on line 95. We will clarify this, since understanding this equation is *fundamental* to the remainder of the paper. As we adopt a *hierarchical* approach, our documents are distributions $(\vec{d}^i \in \Delta^{|T|})$ supported on topics $(\delta_{t_k}, k = 1, \ldots, K$ where *Dirac delta* $(\delta_x)$ at $x$ is a probability distribution only supported on the point $x$). Line 95 says that the distance between two documents is the Wasserstein distance between their distributions over topics, where topics are also distributions, but over words, hence the ground metric is another Wasserstein distance between topics represented as distributions over words. Thus the distance is *hierarchical*.

**Experiments and hyper-parameters**    To answer **R2**'s question on pruning for WMD, we refer to Figures 3 and 5 where WMD-T20 represents WMD truncated to the top 20 words: pruning heuristic *cannot* be efficiently applied to WMD. While it helps with the run-time (see Table 1), it noticeably degrades the performance (see Figure 5).

**R3** asks how to interpret the *results* in Figure 4a. GloVe embeddings used in all of the experiments (including Figure 5) are the *high quality* pre-trained 300d embeddings trained on 6 billion tokes, which can be downloaded online (please see `main.py` file in the code). Figure 4 (a) quantifies sensitivity of different methods to *lower quality* word embeddings. In particular, we trained 200d embeddings with word2vec algorithm using *only* documents of the Reuters dataset (under 300k tokens). Figure 4 (a) shows that lower quality word embeddings significantly degrade performance of the WMD-based methods. Our methods, on the contrary, maintain good performance because they are able to utilize informative topic structure of the Reuters documents, which is independent of the word embeddings quality.

**R3** questions our choice of *20* words for WMD truncation. While the choice of 20 for WMD-T20 is somewhat arbitrary, it is simply made to bring WMD complexity closer to HOTT and show that WMD *cannot* be made computationally efficient using truncation without degrading its performance. WMD-T20 is already slower than HOTT (see Table 1) and performs noticeably worse (see Figure 5); truncating it further will make the performance even worse, while truncating less will quickly lead to impractical run-time, e.g., computing all pairwise WMD distances on the Gutenberg dataset would take $\approx 178$ *days* on a single machine. We are happy to include a sensitivity analysis on the truncation of WMD on one of the smaller datasets.

**R1** asked why we do not compare with TMD. Most importantly, the cubic complexity of the OT linear program remains prohibitive for the number of topics used in TMD, i.e., from Table 1 in TMD paper it can be seen that number of topics they use is only 3-4 times smaller than vocabulary size. We use 70 topics, i.e. over 100 fold vocabulary size reduction on average across datasets. Quantitatively, Figure 3 of the TMD paper suggests that evaluating a kNN classifier on the BBCSPORT (smallest dataset) takes 24h for WMD and 4h for TMD. First, the WMD implementation we use takes 3-4min and, second, HOTT takes only about 40sec. We conclude that HOTT (and even a simply better WMD implementation) is much faster than TMD. Discrepancy in the WMD speed may be due to authors of TMD not fully utilizing sparsity of the documents when calling the linear program solver.

[Meta-Review · NeurIPS 2019]

This paper proposes a distance metric for documents. The proposed solution is to combine latent topics from topic models with the idea of using geometry from word embeddings to compute distances between pairs of documents (as in the WMD metric). First topics are computed, and WMD is performed at the topic level as opposed to the word level. The hypothesis presented is that modeling documents by their representative topics is better for highlighting differences despite the loss in resolution and is similar to how a person would do this task: breaking down each document into concepts, and then comparing the concepts. Since the topics are precomputed for a given corpus, speed up is gained at inference time when computing document similarities. The paper also reasons that since the topics are fewer, one gains interpretability from the proposed topic-level distance measure. The reviewers felt their concerns were addressed by the author response, and there is support for acceptance.